# Alkaline Phosphatase Survey in Pecorino Siciliano PDO Cheese

**DOI:** 10.3390/foods10071648

**Published:** 2021-07-16

**Authors:** Massimo Todaro, Vittorio Lo Presti, Alessandro Macaluso, Maria Alleri, Giuseppe Licitra, Vincenzo Chiofalo

**Affiliations:** 1Dipartimento Scienze Agrarie Alimentari Forestali, University of Palermo, viale delle Scienze, 13, 90128 Palermo, Italy; 2Dipartimento di Scienze Veterinarie, University of Messina, viale dell’Annunziata, 98168 Messina, Italy; vittorio.lopresti@unime.it; 3Dipartimento di Scienze Chimiche, Biologiche, Farmaceutiche ed Ambientali, University of Messina, viale dell’Annunziata, 98166 Messina, Italy; alex.macaluso@virgilio.it (A.M.); vincenzo.chiofalo@unime.it (V.C.); 4Nuovo Consorzio di Tutela del Pecorino Siciliano DOP, via dell’amicizia 26, 91020 Poggioreale (TP), Italy; maria.alleri@unipa.it; 5Consorzio per la Ricerca nel Settore della Filiera Lattiero-Casearia e dell’Agroalimentare, SP 25, 97100 Ragusa, Italy; glicitra@unict.it

**Keywords:** alkaline phosphatase determination, PDO Pecorino Siciliano cheese, raw milk determination

## Abstract

The determination of alkaline phosphatase (ALP) in cheeses has become an official method for controlling cheeses with a protected designation of origin (PDO), all of which use raw milk. PDO cheeses, characterized by high craftsmanship, usually have an uneven quality. However, for these cheeses, it is necessary to establish ALP values so that they can be defined as a raw milk product. In this study, a dataset with Pecorino Siciliano PDO samples was analyzed to determine ALP both at the core and under the rind. The results showed that there was no significant difference between the different zones in Pecorino cheese. A second dataset of 100 pecorino cheese samples determined that ALP was only at the core of the cheese. Moreover, there was a statistically significant difference between the ALP values of cheeses produced with raw milk and those produced with pasteurized milk. Furthermore, according to the temperatures, a wide variability of ALP values was observed in the Pecorino Siciliano PDO samples from the core of the cheeses. This was a result of several under scotta whey cooking methodologies adopted by cheesemakers, which do not permit a clear range. Therefore, further investigation is desirable.

## 1. Introduction

Italy has a long history of cheese production. In Sicily, the largest island in the Mediterranean area, the Phoenician community began the production of cheese. Several archaeological finds have indicated that dairy activity was routinely conducted during the Eneolithic age [1]. Therefore, Pecorino Siciliano PDO is considered the oldest cheese in the EU.

Pecorino Siciliano is a traditional Italian PDO cheese produced throughout Sicily. It is a semi-hard cheese that is manufactured using traditional techniques, i.e., from raw ewe’s milk without any bacterial starters, according to production protocol (GUCE C 170 EUR-Lex-52020XC0518(03)) (Figure 1). Artisanal cheese-making that uses traditional wooden equipment causes a microbiota that is responsible for acidifying curd and maturing cheese that originates from raw milk. This impacts the equipment, the animal rennet, and the transformation of the dairy environment [2]. PDO Pecorino Siciliano cheese is defined as a semi-cooked cheese because the cheese is cooked under hot scotta whey. Scotta whey is a residual product created during the extraction of ricotta cheese; normally, it is used for cooking pecorino cheeses at 74–78 °C for at least two to three hours. The production protocol does not consider where cooking takes place; therefore, some producers use wooden vats and others use steel. Moreover, some producers use either steel or copper boilers. It is well known that the size and type of the container used to cook cheese influences the degree of heat penetration, as does the amount of scotta whey used and its temperature.

To safeguard PDO production from food fraud, it is necessary to develop a control system for raw milk when producing this type of cheese. To date, one of the most widely used analytical systems is the determination of alkaline phosphatase activity (ALP).

ALP is used throughout the world as a marker for the proper pasteurization of milk, because it guarantees hygienic safety [3]. The analysis of ALP in cheese has been described by ISO 11816-2/IDF 155-2 [4]. In the past, this proposed method was not considered appropriate for reflecting the heat treatment of cheese milk in some types of cheese [5]. For cheese, no legal limit has been set, as has been legislated for milk [6]. Further studies, however, have evidenced that processing conditions, texture, size, and high variability can impact the residual activity of ALP and its zonal distribution in cheese [7,8,9]. The temperature at which the curd is heated, as well as the size of the cheese wheel, are the main parameters that influence the residual ALP activity in cheese. Therefore, as well as having a reliable analytical method, there is an urgent need for an appropriate limit for residual ALP activity to characterize cheeses made from pasteurized milk. Based on a study that involved 700 cheese samples from 32 different cheese varieties, Egger et al. [10] proposed a limit for ALP activity (10 mU/g) in cheese developed from pasteurized milk.

Pecorino Siciliano PDO cheese, similar to several artisanal pecorino cheeses produced in southern Italy, is characterized by a wide variability [2,11]. Therefore, at the request of the Protection Consortium, a survey was conducted on Pecorino Siciliano cheese samples subjected to PDO certification in order to define ALP values.

## 2. Materials and Methods

### 2.1. Cheese Production and Sampling

Two datasets of Pecorino Siciliano cheese samples were used in this survey:(1)A total of 98 Pecorino Siciliano cheese samples (0.5 kg), 5–month ripened, were taken. In total, 78 came from 9 dairies in the PDO Protection Consortium. The remaining 20 samples (0.5 kg) were produced in Sicily and came from 10 dairies that declared, on their labels, that the product was “produced with pasteurized milk”. The ALP analysis was detected at the core of the cheese;(2)A total of 36 Pecorino Siciliano cheese samples (0.5 kg), 5–month ripened, came from 6 dairies in the PDO Protection Consortium and from 3 dairies that declared, on their labels, that the product was “produced with pasteurized milk”. The ALP analysis was detected at the core and under the rind of the cheese.

PDO Pecorino Siciliano cheese is cooked under scotta whey, according to the approved protocol (Figure 1). Then, producers choose cooking temperatures, vats, and cooking time on the basis of environmental conditions and their cheese-making experience. This cooking system determines a wide variability in cheese quality and composition. To better understand the results of the conducted survey, cheese temperatures and cooking technology during PDO Pecorino Siciliano production were detected in 9 dairies that belonged to the PDO Protection Consortium (Table 1). Depending on the cooking processes and the temperature detected at the core after cooking stopped, dairies were classified as either weak (t < 47 °C) or severe (t ≥ 47 °C). Moreover, a further class was formed and called “mixed”, which included cheeses produced by cheesemakers who apply a severe cooking method and lower cooking temperatures (Table 1).

### 2.2. Analysis of Alkaline Phosphatase with Fluorophos^®^

Cheese samples were planned by the PDO Protection Consortium and the Corfilac Consortium. From each, a slice of cheese (0.5 kg) was taken, vacuum-packed, and transferred to a laboratory at the University of Messina. Samples were refrigerated at 5 ± 2 °C. The samples were weighed, and the size of each slice was measured.

All samples were analyzed at the core. Altogether, 98 cheeses samples were taken from the central part of the slice (core), and the distance from the core to the surface was measured (Figure 2). The size of the Pecorino wheels, about 7 kg in weight, gave a height of 17 cm and a diameter of 25 cm.

ALP analyses were conducted at the core and under the rind in 9 cheeses, 6 of which were made with raw milk and 3 of which were made with pasteurized milk. In total, 98 cheeses were sampled. For each cheese, a 1 cm × 1 cm section was taken: 2 at the core and 2 under the rind. The distance of each sample taken at core was higher than 6 cm from the core to the surface, while distance of each sample taken under the rind was approximately 1 cm under the rind (Figure 3). Each portion was finely ground and analyzed, as described below. Cheese samples were analyzed according to the ISO11816-2/IDF 155-2 (second edition 2016.08.15), apart from some optimizations.

Analyses were carried out using a Fluorophos^®^ ALP test system (Advanced Instruments Inc., Norwood, MA, USA) and calibrated with a Calibrator Set FLA250 (Fluorophos^®^ Advanced Instruments Inc., Norwood MA, USA) using a pasteurized pecorino. This was undertaken in order to consider the matrix effect.

A cheese extraction buffer FLA005 (Fluorophos^®^ Advanced Instruments Inc., Norwood MA, USA) was used to extract the cheese. An aliquot in the cheese grinder was weighed to the nearest 1 mg in a 15 mL conical test tube. Next, 5 mL of the cheese buffer was added and mixed with a homogenizer (Ultra-Turrax^®^ T 25 basic IKA^®^-WERKE, Janke and Kunkel-Str. 10 79219 Staufen, Germany) for 60 s to obtain a completely homogenous dispersion. Another 5 mL Cheese Extraction Buffer was used to rinse the homogenizer and this emulsion was added to the previous extract.

The final sample was centrifuged at 1000 g/min at 4 °C for 10 min. The upper phase was collected in a clean tube, wherein 25–75 μL were withdrawn for analysis, according to ISO 11816-2/IDF 155-2 [4].

The instrumental results were converted into dilution factors and expressed in mU/g. Before each analysis, the instrument was checked according to the manufacturer’s instructions and the ISO ISO11816-2/IDF 155-2. This was accomplished by performing tests with a Daily Instrument Control^®^ FLA 280 and Phospha Check Pasteurization Control^®^ FLA260 (Fluorophos^®^ Advanced Instruments Inc., Norwood, MA, USA).

### 2.3. Statistical Analysis

#### 2.3.1. First Data Set

Here, a total of 36 cheese samples were statistically analyzed. We detected a lack of normal distribution for ALP values. As such, a logarithm transformation of the original data was applied. The effect that the type of sample (core or under-rind) had on the analysis was detected using ANOVA analysis. Using a GLM procedure [12], the following model was implemented:ALP_jlk_ = µ + Sample_j_ + Milk_l_ + (Sample × Milk)j_l_ + ε_jlk_(1)
where ALP_ijk_ is the ALP logarithm value; Sample_j_ is the part of the cheese taken, wherein j indicates the core or under the rind; Milk_j_ is the type of milk used for cheesemaking, wherein j indicates either raw or pasteurized. Means were compared using the Student’s *t*-test. Moreover, *p* values less than 0.05 were considered to be statistically significant.

#### 2.3.2. Second Data Set

In this study, at least 98 cheese samples were analyzed, and simple statistics were calculated. We detected a lack of a normal distribution for ALP values. As such, we applied a logarithm transformation to the original data. The effect of under scotta whey cheese cooking on ALP logarithmic values was detected using ANOVA analysis. Using a GLM procedure [12], the following model was implemented:ALP_jk_ = µ + Cooking_j_ + ε_jk_(2)
where ALP_ijk_ is the ALP logarithm value and Cooking_j_ is the classification of under scotta whey cheese cooking, wherein j indicates severe, weak, mixed, or pasteurized milk. Means were compared using the Student’s *t*-test. Moreover, *p* values less than 0.05 were considered to be statistically significant.

## 3. Results and Discussion

Several studies reported that cheese size impacts the residual activity of ALP and its zonal distribution [6,7,8,9,10]. Additionally, several authors agree that the temperature at which the curd is heated, and the size of the cheese wheel, are the main parameters that influence residual ALP activity in cheese [6,10].

The particular cooking technology applied to PDO Pecorino Siciliano cheese requires in-depth study in order to determine ALP values in different areas of the cheese. Table 2 shows the statistical analyses of 36 Pecorino Siciliano cheese samples. As expected, ALP values detected on cheeses produced with raw milk showed significantly higher values than cheeses produced with pasteurized milk. The effect of the cheese area was not statistically significant for PDO cheeses produced with raw milk, while significant differences in ALP values were found between the core and under-rind areas in cheeses produced with pasteurized milk. In accordance with Egger et al. [10], ALP values detected at the core were significantly lower than ALP values under the rind, probably due to a higher temperature at the core of the cheese and slower cooling. The cheese area (core or under-rind) in PDO Pecorino Siciliano cheeses was found to have no significant influence on the ALP value, probably due to the wide variability that characterizes these cheeses. This is because they are produced in an artisanal way, wherein the cheesemaker is able to modify the quality of the cheese while respecting PDO protocol. The effect of the cooking process (i.e., severe, lack, and mixed) was tested in the statistical model, but it was not significant and, thus, was deleted.

Table 3 shows the simple statistics of ALP real values detected in Pecorino Siciliano cheese samples, which were produced by nine dairies that belong to the PDO Protection Consortium. In accordance with the classification proposed in Table 1, Table 3 shows two groups of results. Cheeses with a severe process showed a mean range between 26.5 and 909.1 mU/g, while cheeses cooked with a weak process showed a mean range between 1952.0 and 3148.6 mU/g. Even within the same cooking class, the variability between different dairies was wide. This is likely due to the different cooking systems practiced by different cheesemakers, who change the cooking system according to their daily needs or environmental temperatures. For this reason, a third group of cheeses was created (i.e., mixed). This group was produced by dairies that adapted a severe cooking system but functioned in different environmental conditions and used lower cooking temperatures. This cooking class presented a mean value similar to the weak category, i.e., equal to 1699.6 mU/g, but with a wide standard deviation, 1152.6 mU/g.

Therefore, even when the same dairy is used during cheese production, different values of alkaline phosphatase can be recorded.

Pecorino Siciliano cheese samples that report on the lables “pasteurized milk product” present a mean value of 12.3 mU/g with a range between 5.8 and 23.9 mU/g. The ALP mean value of these cheeses is lower than Pecorino Siciliano PDO cheeses. This is true even for those produced under severe cooking conditions.

In this study, we reported the statistical analysis carried out on ALP logarithmic values (Table 4). The statistical model explains that 89% of variability (R^2^) resulted in a statistically significant factor (*p* < 0.001). The logarithmic least square means (LSM) were statistically different between cooking classes. The LSM of cheese samples produced with pasteurized milk was 1.048, which corresponded to 15.7 mU/g. This was statistically lower than raw milk cheeses cooked under a weak (3.308; *p* < 0.001), severe (1.612; *p* < 0.001), or mixed cooking process (3.128; *p* < 0.001). For cheeses produced with raw milk, ALP values were classified as weak and were significantly higher than those in the severe cooking class. No differences were found with respect to the mixed class. ALP values in the severe cooking class were significantly lower than those in either the weak or mixed classes.

The above results appear to conflict with those reported in the literature [10]. However, few papers have analyzed the value of phosphatase on cheeses and, in particular, on traditional cheeses, because this method has only recently been validated [6,10,13].

Different cooking conditions determine the variability of alkaline phosphatase activity, such as the temperature of scotta whey before cooking, cooking time, the ratio between the liters of scotta whey used and the kg weight of the cheese. Severe cooking conditions for PDO Pecorino Siciliano cheeses determined low ALP values, which could potentially be misattributed to the thermal treatment of raw milk. In fact, in accordance with Egger et al. [10], these cheeses could have been produced by thermized milk. However, based on the control of cheese manufacturing procedures, the temperatures used in these dairies guarantees a more rigorous application of PDO protocol, ensuring that these values are sufficiently reliable.

A variability of ALP values was detected in Pecorino Siciliano cheese samples, particularly those produced with pasteurized milk. Several samples presented ALP values higher than 10 mU/g, a threshold reported by other authors [6,9,10]. These values discriminated against cheese produced with raw or pasteurized milk. A probable justification for this is that the production processes of these pecorino cheeses were not checked by us, and the sampling occurred on the basis of what was declared by the producer, who may have used different pasteurization temperatures.

## 4. Conclusions

In conclusion, we determined that the production of PDO Pecorino Siciliano cheese is characterized by a strong craftsmanship that indicates a wide variability between dairies. Different cooking systems that used scotta whey at various times and temperatures, and in different types of vats, showed a wide variability of ALP values and an unclear demarcation between Pecorino Siciliano cheeses produced with raw milk or pasteurized milk.

The presence of low ALP values detected in PDO Pecorino Siciliano samples that underwent severe cooking temperatures at the core of the cheese (i.e., above 47 °C), but were produced with raw milk, fall within the range of ALP values detected in pasteurized milk cheeses. The overlap of these values suggests that caution should be used when applying alkaline phosphatase as a control tool on raw or pasteurized milk cheeses. According to Clawin-Rädecker et al. [6], further scientific studies are necessary to establish whether there are threshold values capable of discriminating against the type of milk (raw or pasteurized) used for producing PDO Pecorino Siciliano.

## Figures and Tables

**Figure 1 foods-10-01648-f001:**
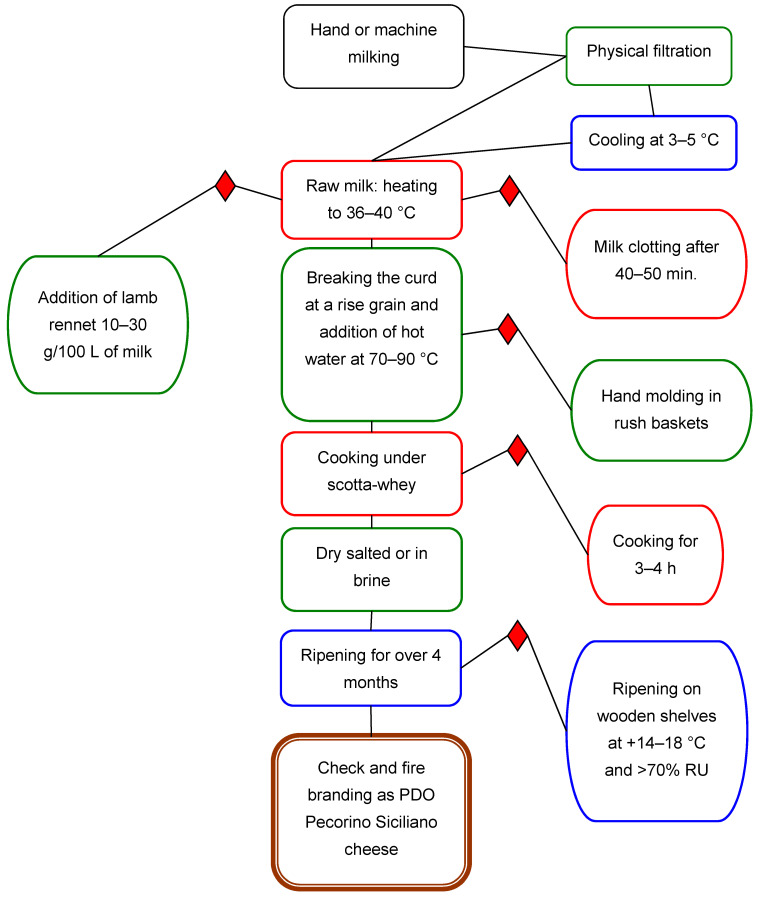
Flow chart of PDO Pecorino Siciliano cheese.

**Figure 2 foods-10-01648-f002:**
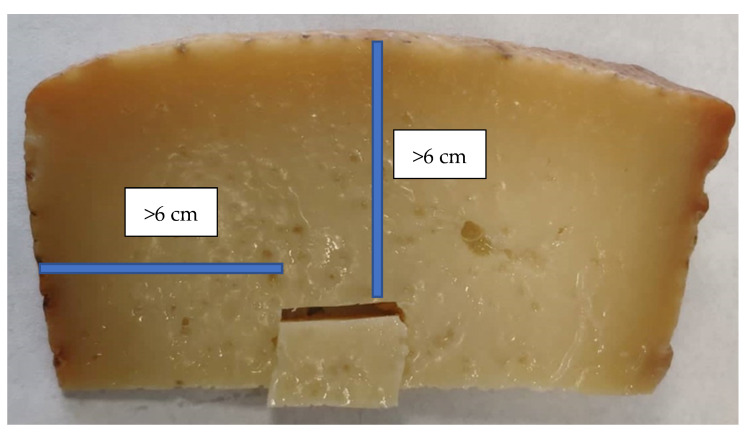
Cheese sample taken at core during the ALP analysis.

**Figure 3 foods-10-01648-f003:**
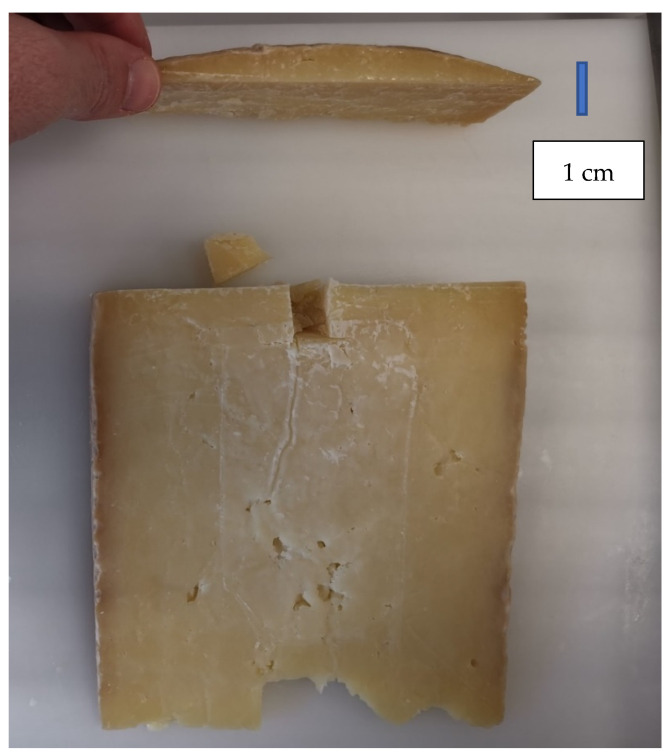
Cheese sample taken under the rind during the ALP analysis.

**Table 1 foods-10-01648-t001:** Cooking technology detected in dairies belonging to the PDO Protection Consortium for Pecorino Siciliano cheese.

Dairy	Temperature of Cheese before Cooking (°C)	Temperature of Scotta Whey before Cooking (°C)	Ratio between Scotta Whey Liters and kg of Cooked Cheese (L/kg)	Temperature of Cheese after Cooking (°C)	Temperature of Scotta Whey after Cooking (°C)	Cooking Time (h)	Cooking Classes
A	39.5	75.0	5	54.5	57.5	4.5	severe
B	43.0	74.0	4	52.0	54.0	4	severe
C	39.7	74.0	3.5	50.0	59.0	4.5	severe
D	42.0	77.0	4	47.0	58.0	3	severe
E	38.5	78.0	3.5	47.5	65.0	3.5	severe
F	38.8	72.5	3	49.5	59.2	2	severe
H	34,5	73.5	2	44.7	52.2	3	weak
I	41.0	77.0	3	44.0	44.0	3	weak
L	37.5	75.0	3	45.5	52.0	3	weak
Other PDO cheesemaker	38.0	74.7	3	45.0	34.5	2.8	mixed

**Table 2 foods-10-01648-t002:** ALP values in different zones of the cheese slice.

Milk	Sample	ALP (log 10)	Standard Error	Probability *p*<		
				Milk	Sample	Milk × Sample
Raw	Core	3.211 A	0.126	0.001	0.069	0.028
	Under-rind	3.147 A	0.126			
Pasteurized	Core	0.890 C	0.178			
	Under-rind	1.536 B	0.178			

On the column: A, B, C: *p* ≤ 0.01.

**Table 3 foods-10-01648-t003:** ALP values in dairies belonging to the PDO Protection Consortium.

Dairy	Cooking Classes	Temperature of Cheese after Cooking (°C)	*n*.	Mean (mU/g)	SD (mU/g)	Max (mU/g)	Min (mU/g)
A	severe	54.5	5	26.5	14.9	47.7	13.0
B	severe	52.0	5	539.3	695.8	1756.1	11.3
C	severe	50.0	10	88.1	111.0	298.5	13.7
D	severe	47.0	4	839.0	1614.1	3714.8	25.2
E	severe	47.5	5	86.9	161.2	375.0	7.8
F	Severe	49.5	3	909.1	1198.8	2831.9	37.2
H	weak	44.7	12	2992.8	1533.8	5543.4	1433.3
I	weak	44.0	20	1952.0	1363.9	4150.2	512.6
L	weak	45.5	6	3148.6	382.7	3781.2	2682.4
Other PDO cheesemaker	mixed	45.0	8	1699.6	1152.6	3714.8	537.0
From the market	Pasteurized milk	40.0	20	12.3	5.9	23.9	5.8

**Table 4 foods-10-01648-t004:** Least square means (LSM) of ALP logarithmic values.

Cooking Classes	ALP (log 10)	Standard Error	ALP (mU/g) *
Weak	3.308 A	0.055	2032.3
Severe	1.612 B	0.060	40.9
Mixed	3.128 A	0.120	1342.8
Pasteurized milk	1.048 C	0.076	15.7
Cooking classes	*p* < 0.001		R^2^ = 0.89

On the column: A, B, C: *p* ≤ 0.01; * these values were calculated as the inverse of logarithmic LSM.

## Data Availability

The data presented in this study are available on request from the corresponding author.

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
