# Peer review of "Alkaline Phosphatase Survey in Pecorino Siciliano PDO Cheese"

_foods, 2021, doi:10.3390/foods10071648_

Round 1

Reviewer 1 Report

The article discusses an important issue, however, it has some flaws that need to be corrected.

  • A linguistic review of the paper should be performed. It is probably such an issue that the Authors refer to Pecorino Siciliano as being made "from raw ewe’s milk" instead of whole sheep's milk.
  • Scientific soundness of a manuscript strongly depends upon its literature review. In this case an important recent EFSA scientific report should have been included. The Authors should also outline other potential analytical tools for verifying the manufacturing process (i.e. use of unpasteurized milk).
  • Description belonging to Table 1 is not clear: what is the difference between "Other PDO cheesemaker" process and the "weak" processes?
  • Storage time from the production (ripening period) and in the laboratory can both have an impact on the results. Please, specify these!
  • The Authors state that "...the size of the cheese wheel, are the main parameters that influence the residual ALP activity in cheese", however, they didn't report any data referring to this dimension
  • Measurement units should be given in Table 1
  • More exact definition of sampling place should be reported (distance from rind for samples taken from the zone under the rind, distance from perimeter for samples taken from the core, for both categories distance from bottom and upper part of the cheese wheel)
  • L 116-118: time of homogenization should be given and description of the repeated extraction procedure (?) should be given more clearly.
  • L132: each member of the equation should be explained
  • Why was Student test applied for detection of differences instead of ANOVA test?
  • Was any outlier deetction performed? See e.g. sample G (Table 23), which is probably an outlier, so it shouldn't be included in mean value for the "weak" process
  • Is there any explanation for sample G (Table 3)? Significant differences should be given for ALP values in Table 3.
  • L222-223, L223-224: please, clarify these sentences!
  • Were ALP values referred to in L223-224 really "superimposable"? Based on the values reported, this is apparently not the case. 

Reviewer 2 Report

PDO Pecorino Siciliano cheese is a local product made in a traditional way according to the recipes passed from one generation to another. However, even in the place of its origin (Italy),  there are often counterfeit cheese made from cow's milk. Therefore, there is a significant need to protect this traditional product.

The variability of the quality of the cheese obtained is very large, because it depends on many external factors indicated by the authors, as well as on the craftsmanship of the cheese maker themselves, who works in accordance with a long-term tradition. Hence, the obtained results may be different. During Pecorino production, “Scotta,” which is the main by-product of Ricotta production process, is used. It is a pity that the authors have not described in detail what is the purpose of this process and how it affects the quality of the cheese.

The manuscript is missing the information on the novelty behind the production technology/ analysis / additives used. Ultimately, cheese is a food product, therefore, it is important to understand how the consumers perceive it. Does the presence of ALP affect the palatability of the cheese? There is a lack of a sensory analysis.

In my opinion, the authors described only one type of analysis performed on a traditional local product (ALP). This information is very limited and does not give a complete picture of the quality of the cheese produced. If the authors have more data, I would advise to enrich the article.

Round 2

Reviewer 1 Report

There are still some minorerrors that have not benn corrected:

  • EFSA reference is not given correctly
  • significant differences in Table 3 are still missing (annotation with letters)
  • some minor linguistical errors are still occurring (see e.g. L 171, L206-207)

Reviewer 2 Report

The authors considered my comments to be unfounded. Therefore, my assessment of the manuscript remains as I originally described.
